# Plant Executor Genes

**DOI:** 10.3390/ijms23031524

**Published:** 2022-01-28

**Authors:** Zhiyuan Ji, Wei Guo, Xifeng Chen, Chunlian Wang, Kaijun Zhao

**Affiliations:** 1National Key Facility for Crop Gene Resources and Genetic Improvement (NFCRI), Institute of Crop Sciences, Chinese Academy of Agriculture Sciences (CAAS), Beijing 100081, China; Wangchunlian@caas.cn; 2College of Chemistry and Life Sciences, Zhejiang Normal University, Jinhua 321004, China; weiguo817@zjnu.cn (W.G.); xfchen@zjnu.cn (X.C.)

**Keywords:** executor gene, TALE, *R* gene, innate immunity, plant resistance

## Abstract

Executor (*E*) genes comprise a new type of plant resistance (*R*) genes, identified from host–*Xanthomonas* interactions. The *Xanthomonas*-secreted transcription activation-like effectors (TALEs) usually function as major virulence factors, which activate the expression of the so-called “susceptibility” (*S*) genes for disease development. This activation is achieved via the binding of the TALEs to the effector-binding element (EBE) in the *S* gene promoter. However, host plants have evolved EBEs in the promoters of some otherwise silent *R* genes, whose expression directly causes a host cell death that is characterized by a hypersensitive response (HR). Such *R* genes are called *E* genes because they trap the pathogen TALEs in order to activate expression, and the resulting HR prevents pathogen growth and disease development. Currently, deploying *E* gene resistance is becoming a major component in disease resistance breeding, especially for rice bacterial blight resistance. Currently, the biochemical mechanisms, or the working pathways of the E proteins, are still fuzzy. There is no significant nucleotide sequence homology among *E* genes, although E proteins share some structural motifs that are probably associated with the signal transduction in the effector-triggered immunity. Here, we summarize the current knowledge regarding TALE-type avirulence proteins, *E* gene activation, the E protein structural traits, and the classification of *E* genes, in order to sharpen our understanding of the plant *E* genes.

## 1. Introduction

Over the course of their lifetimes, plants are exposed to microorganisms in the environment and are faced with continuous hostilities from pathogenic microbes [1]. To cope with the invasion of a wide variety of pathogens, plants have established multiple layers of immune responses [1,2]. Proteins encoded by resistance (*R*) genes are fundamental to the innate immunity of plants because they perceive pathogen effectors and switch on the immune response through various mechanisms [3]. Most of the plant *R* genes are the NLR type because they encode nucleotide-binding leucine-rich (NLR) repeat receptor proteins. However, several other plant *R* genes encode executor proteins (Table 1), which directly kill the plant cells infected by the pathogen and that can, therefore, be called “executor” (*E*) genes. The known *E* genes were merely identified from rice and pepper [4,5,6,7,8,9,10,11]. The *E* genes can trap a unique class of bacterial effectors through the specific sequences located in their promoters, termed the “effector-binding elements” (EBEs), and can initiate expression to stimulate the defense response [12,13,14]. The activated expression of *E* genes always leads to a hypersensitive response (HR) in the host for the prevention of pathogen growth and disease development.

TALEs (transcription activation-like effectors) in pathogens are the natural triggers for *E*-gene-mediated disease resistance. They exist only in plant phytopathogenic *Xanthomonas* and *Ralstonia*, which cause diseases in many plant species, including important crops [15,16]. TALEs belong to a family of structurally conserved type III effectors and function as eukaryotic transcription factors in inducing gene expression. Besides the avirulence functions, they are also critical for pathogen fitness and virulence, and they sometimes even determine the virulence difference for bacteria isolates [14,17,18]. The original intention, for pathogens to deliver TALEs into plant cells, is to enhance the host susceptibility by manipulating the expression of susceptible (*S*) genes, such as transporter genes (*SWEET* and *SULTRP*), transcription factor genes (*OsTFX1* and *OsTFIIA**γ1*), etc. [14,18,19,20,21].

On the other hand, plants have evolved two major strategies to defend themselves against attacks from TALEs. NLR proteins, such as XA1 and its allelic R members, can broadly recognize TALEs and can initiate an immune response via a direct or indirect protein–protein interaction [22,23,24,25]. Unfortunately, the broad-spectrum resistance mediated by XA1 and its homologues is defeated by the new virulence factors called iTALEs (interfering TALEs), which are prevalent in *Xanthomonas oryzae* pv. *oryzae (Xoo)* and *Xanthomonas oryzae* pv. *oryzicola (Xoc)* isolates. Currently, the mechanisms under either the TALE-triggered resistance or the iTALE-mediated suppression of the NLR-mediated resistance are issues to be addressed for the development of methods to recover the NLR protein resistance against *Xoo* and *Xoc* [26]. *E* genes are another route for plants to counter attacks from TALEs [14,18,21]. An *E* gene activates effector-triggered immunity (ETI) via the trapping of the cognate TALE. It is worth noting that some TALEs, such as AvrXa7 and AvrBs3, possess dual functions (virulence and avirulence) from the continuous evolutionary battles between plants and pathogens. However, these double-faced TALEs can still induce a disease-resistance phenotype when resistant and susceptible targets are simultaneously induced [5,9,10,11]. 

Generally, *E* genes are “silenced” in the absence of a pathogen attack, and the induction of *E* gene expression can lead to programmed cell death (PCD) in multiple plant and mammalian cells, implying that the launched immune response is activated through some conserved mechanism [7]. However, the detailed mechanism for *E*-gene-mediated immunity is still ambiguous. *E* gene family members encode short proteins (less than 350 aa) with no significant sequence similarity. Zhang et al. have classified *E* genes into two groups on the basis of the characteristics of the encoded products [27]. Bs3 is the sole member of the Group 1 E protein (G1EP), and it is homologous with the flavin mono-oxygenases (FMOs) of the *Arabidopsis* YUCCA (YUC) family, which is in a critical position for systemic acquired resistance (SAR) [28,29,30]. The members in the Group 2 E proteins (G2EPs) have some common features (Figure 1): (1) G2EPs have multiple potential hydrophobic membrane-spanning domains (22~23 aa), with imperfect L(I)-X(4-9)-L(I) motifs, probably forming ion (probable Ca^2+^) channels in membranes; (2) The C-terminal domain contains the ED motif, or the acidic amino acid residues (Glu and Asp), required for host resistance and HR induction in nonhosts, for example, *Nicotiana benthamiana*; (3) The known G2EPs, except for XA27, localize to the nuclear envelope and the endoplasmic reticulum (ER) membrane, and are supposed to release Ca^2+^ from stores into the cytoplasm [7,31]. In this review, we summarize the current knowledge regarding TALE-type avirulence proteins, the E protein structural traits, genetically engineered *E* genes, and the possible mechanism for signal transduction in the *E*-gene-mediated defense response.

## 2. Interaction between *E* Genes and Cognate Avirulence Proteins

The interaction between an *E* gene and a TALE is a gene-for-gene relationship. TALEs have several well-characterized domains, which are conserved in the family and are essential for inducing the expression of their target genes [32,33,34]. A typical TALE has type III secretion and translocation signals in the N-terminal to direct the effector protein to enter host cells via the bacterial type III secretion system (T3SS). In the plant cell, the host cytoplasm/nuclear shuttle proteins, OsImpα1a and OsImpα1b, transport the TALEs into the nucleus by binding to the nuclear localization signal (NLS) at the end of the C-terminal of the effector proteins [34]. Then, the TALEs are guided into the plant nucleus, and the central repeat region (CRR) binds with the specific EBE sequences of the target gene promoters. The main differences between the TALE members are the numbers and arrangements of the repeats in the CRR, which are composed of the tandem repeats of a 33-to-35 amino acid motif. The basic repeat motif of the CRR is nearly identical, apart from the repeat-variable diresidues (RVDs) at the 12th and 13th amino acids. The sequence of the RVDs determine the specificity of a TALE, and each repeat of a TALE finally interacts with a specific nucleotide in the EBE [35,36]. A single TALE can affect the race classification of the pathogen isolates when its target is important in the host–pathogen response [20].

To successfully active the expression of the target genes, TALEs still need to hijack the host basal transcription factors, TFIIAα and TFIIAγ, via the transcription factor binding (TFB) region [37,38,39]. Rice have evolved several precise disease resistance mechanisms to evade the operations of TALEs [39]. *xa5* is the resistant allele of *Xa5*/*OsTFIIA**γ5*, and it encodes a variant with a valine-to-glutamic acid change (V39E) [40]. The single amino acid mutation attenuates the interaction between TALEs and OsTFIIAγ5, and it leads to a reduction in the TALE-dependent induction of the downstream target genes. *xa5* confers rice resistance to various strains of *Xanthomonas oryzae* (*Xoo* and *Xoc*). The low binding affinities of TALEs and *xa5* also interfere with the resistance of *E* genes, such as *Xa23* [39]. However, TALEs can recruit OsTFIIAγ1, another OsTFIIAγ protein, to compensate for the absence of *Xa5* [41]. In addition, plants also apply another strategy to prevent the TALE binding by disrupting the EBE sites of TALEs, such as the resistance conferred by *xa13* and *xa25* [21]. 

*E*-gene-mediated resistance is a form of the plant’s active defense against TALEs. Compared with the recessive gene, the dominant character makes *E* genes easier to breed into receptor cultivars of crops. Six pairs of *E* genes, and their corresponding avirulence (*avr*) genes, have been cloned from host–*Xanthomonas* interactions [27]. The disease-resistant spectrum of *E* genes depends on the recognized TALEs. *Xa23* has the broadest spectrum of resistance among all the bacterial blight *R* genes because *avrXa23* exists in all of the tested *Xoo* strains [8]. Two TALE members, AvrBs3 and AvrXa7, are recognized by *Bs3* and *Xa7*, and can target *up20* and *OsSWEET14* in pepper and rice, respectively [42,43]. Both of the inductions of *up20* and *OsSWEET14* contribute greatly to enhance the susceptibility to pathogen infection. However, the connection between the evolution of the *E* gene and the *S* gene has not been elucidated clearly.

## 3. Executor Genes in Rice–*Xanthomonas oryzae* Interaction

The rice–*Xanthomonas oryzae* (*Xo*) interaction system is an ideal model for probing the role of *E* genes in the innate immunity of plants [21,44]. *Xoo* and *Xoc* are two closely related pathovars, and both of them utilize TALEs as key virulence factors to subvert the host immune system [18,21,45]. Nevertheless, the TALEs from the two pathogens manipulate the innate immunity of rice by targeting different types of *S* genes. *E* genes make different contributions to the defense responses against *Xoo* and *Xoc*. Four *E* genes, *Xa7*, *Xa10*, *Xa23*, and *Xa27,* have been cloned from rice and have been widely used for the improvement of bacterial blight resistance in breeding [4,7,8,9,10,11]. Although these *E* gene members were classified into the same group, G2EP [27], significant differences exist for HR induction in nonhost tobacco. The transient expression of *E* genes from rice can induce clear HR in *N. benthamiana*, but not in *Xa27*. A possible explanation is that the HR induction in tobacco is associated with the localization of E proteins. The XA27 is an apoplastic protein that depends on an N-terminal signal–anchor-like sequence to localize to the apoplast, which is required for resistance to *Xoo*. Other G2EP members (XA7, XA10, and XA23) from rice are ER-located proteins without signal–anchor sequences [4,7,9,10,46]. This suggests that TALEs use the same action mode to activate the *E* gene, but the resistance pathways and mechanisms for *Xa27* and other rice *E* genes are not identical. Despite the low sequence similarity between the E proteins, XA7, XA10, and XA23 share the same structure features. We presume that they form a calcium-permeable cation channel (similar to the ZAR1 resistosome), or that they interact with the existing ion pumps on the ER membrane to release calcium ions into the cytoplasm in order to trigger immunity and cell death [7,47].

*Xoc* harbors the largest group of TALEs in the phytopathogen bacteria, usually more than 20 TALE members per strain. Unfortunately, no effective *E* gene against *Xoc* has been identified from the natural germplasm resources. Hummel et al. report that Tal2a_BLS256_ elicits dose-dependent resistance in rice, but it is insufficient to produce a disease-resistant phenotype or HR in the *Xoc* background [48]. The believable explanation is that there are complicated interplays between effector-triggered immunity (ETI) and effector-triggered susceptibility (ETS) in rice–*Xoc* interactions. The avirulence of TALEs is masked by the suppression of *Xoc* [49], probably from some virulent TALEs.

## 4. Executor Genes in Pepper–*Xanthomonas campestris* pv. *vesicatoria* Interaction

*E* genes have also been discovered from another plant–pathogen interaction system, between pepper (*Capsicum* L.) and *Xanthomonas campestris* pv. *vesicatoria* (*Xcv*). *Xcv* is the causal agent of bacterial spot disease in pepper (*Capsicum* subspecies (spp.)) and tomato (*Lycopersicon* spp.) [50]. TALEs contribute significantly to the fitness for *Xcv*, but they are not major virulence determinants, as they are in *Xanthomonas oryzae*.

*Bs3* is an exceptional member of Group 1 *E* genes. AvrBs3, the matching avirulence protein for *Bs3*, can induce cell hypertrophy in susceptible plants, which might facilitate bacterial spreading [51]. Besides *Bs3*, AvrBs3 can also be recognized by another NLR protein, Bs4, but the molecular mechanism is unknown [52]. Bs3 is highly related to AtYUC8, of the AtYUC protein family, with characteristics of the FAD- and NADPH-binding sites. The mechanism required in order for Bs3 to initiate the immune response is relatively clear. The credible hypothesis is that the Bs3-mediated immune response relies on some established pathways contributing to SA and Pip accumulation, and these established immune signaling pathways are probably shared with other R proteins. Many comparative experiments also indicate that the FAD- and NADPH-binding sites, unique sequence and protein localizations, have a profound impact on Bs3 signal transduction [28]. AvrHah1, a single TALE member from another pathogen, *Xanthomonas gardneri*, is 87% identical in sequence to AvrBs3, but it also activates Bs3 resistance in pepper [53]. The results show that *Bs3* can overcome small mismatches in the EBE to trap the cognate TALEs [54].

*Bs4C-R*, originating from *C. pubescens* PI 235047 (*CpBs4C-R*), is an *E* gene member of Group 2, and it contains an EBE trap for AvrBs4 [6]. AvrBs4 interacts with *CpBs4C-R* in a typical TALE–DNA action mode, induces catalase crystals in peroxisomes, and triggers a HR in resistant pepper genotypes. CpBs4C-R and homologues from pepper (CaBs4C and CpBs4C-S) have negligible differences at the amino acid level, and show closer structural homology with XA7, XA10, and XA23 in the G2EPs [31]. Additionally, *Bs4C* genes can not only induce cell death in pepper and *N*. *benthamiana*, but they can also switch on the immune response in rice, as in the three rice *E* gene members, after inducement by the suitable TALEs [31]. Therefore, we infer that the G2Eps from rice and pepper, except for XA27, possess key similarities in their biochemical properties and underlying molecular mechanisms for the induction of the defense response. 

## 5. Genetically Engineered *E* Genes for Combating *Xanthomonas* Diseases

The molecular basis for TALE–DNA interactions has been well elucidated and it promotes control strategies for multiple plant diseases. The disruption of the EBE sites in susceptible targets, through TALEN- and CRISPR-based genome editing, was proven effective and is now widely applied in order to endow the host with broad-spectrum resistance to *Xanthomonas* pathogens [55,56,57,58,59]. 

On the contrary, another common method that has been exploited is the integration of designed EBEs into the promoter region of an *E* gene (or its alleles) in order to expand the resistance spectrum, or to obtain additional resistance to other pathogens by targeting the desirable TALEs [60,61,62]. A single naturally evolved *E* gene basically recognizes one TALE, and it often exhibits race-specific resistance. Multiple tandem EBEs not only confer *E*-gene (or its alleles) resistance to strains of different races, but also to other TALE-carrying bacterial pathogens of different host plants. However, *E* gene transgenic plants were frequently found to show stress-related phenotypes, including lesion mimics and the retardation of growth and development. It has been explained that the *E* gene was expressed constitutively and when using conventional transgenic approaches [9,10,11]. CRISPR-Cas9-mediated homology-directed repair (HdR) can precisely manufacture the engineered promoters for trapping the corresponding TALEs, and it can successfully dodge the damages from leaky *E* gene expression [63]. Therefore, there is great value in identifying new *E* genes and their alleles from different plant species. Furthermore, suitable pathogen-responsive *cis*-elements can be introduced into the promoter at the native locus in order to direct *E* gene expression, which could confer multipathogenic resistances without inducing excessive stress responses. 

To obtain broad-spectrum and durable resistance for *Xanthomonas* diseases, the EBEs targeting the important and widespread TALE members were preferentially integrated into the engineered *E* genes. Investigations into the TALE distributions in pathogens are essential to the design of ideal recognition sequences. The conserved TALEs of *Xanthomonas campestris* pv. *campestris* have some differences in RVDs, but they bind to the same EBE site in the *ERF121* promoter, which is ideal for engineering *E* genes against the black rot of crucifers [64]. Five conserved TALEs and their candidate targets were identified in the projects for the *Xoc* whole-genome sequencing, and the novel recognition specificities will display enormous value in BLS controlling [65].

## 6. Discussion

*E* genes, a novel paradigm of *R* genes, have an exclusive role in the innate immunity of plants, and they only function when pathogen effectors are present [66]. Theoretically, the activation mode of an *E* gene is ideal for balancing the immunity and yields in crop breeding, an area for which the *E* gene family members have bright prospects [67]. Actually, in our conventional breeding programs, the introduction of an *E* gene, *Xa23,* was also found to have no negative effect on the agronomic traits of rice. However, the leaky expression of *E* genes, or their functional derivatives, was found to cause lethal phenotypes in transgenic plants [7,9,10,62,63]. We speculate that some transcriptional repressive elements could be located upstream of the *E*-gene-coding region, at the native locus, to keep the background gene expression at a favorable level.

*E* genes seem to be special *R* genes against *Xanthmonas* diseases. More recently, a new barely leaf rust *R* gene, *Rph3*, was cloned and assumed to be a new *E* gene member that encodes a small protein with multiple transmembrane domains and induces host cell death in a similar manner to the G2EPs [68]. The isolation of new putative members of the *E* gene family from the barley–*Puccinia hordei* interaction system suggests that similar host defense mechanisms probably evolved independently in different plant–pathogen interaction systems, and TALEs might not be a unique type of avirulence factor for *E* genes. *E* genes should make up a higher proportion in plant *R* genes, which is now 6/314 in the cloned *R* genes [3]. Up until now, most of the identified E proteins, including Rph3, are consistent in their structural characteristics, but highly variable in their amino acid sequences [9,69]. For example, we found that the XA23 variants have a different number of amino acids in the C-terminal ED motif (our unpublished data). The variation did not influence the disease-resistance performance of E proteins in plants. We hypothesize that this could be associated with the intensity of the resistance response, probably to reduce the potential toxic effects in the given backgrounds. Interestingly, we also observed that the transient expression of some *E* genes, such as *Xa23*, failed to induce a HR in some plant species, such as *Arabidopsis*. This result indicates that the signal transduction pathways for *E*-gene-mediated immunity probably differ among plant species.

Although all E proteins can trigger cell death pathways, they have obvious differences in the subcellular localization. Bs3, the most extensively investigated E protein member, can initiate ETI, irrespective of whether it is localized in the nucleus or the cytoplasm [28]. XA27 is an apoplastic protein, and localization is required for its function in conferring disease resistance [46], while other members of the G2EPs, which include Bs-4CR, XA7, XA10, and XA23, are ER-located proteins, which might have an internal signal–anchor sequence [7,11,31]. The correlation between the E protein subcellular localization and the signal transduction pathway remains largely unknown. *E*-gene-mediated immunity relies on some conserved, but not identical, signaling pathways, which are important components of the innate immune system of plants.

The functions of the E protein and the signal transduction in the defense response are the pivotal issues for plant immunity. Recent breakthrough studies have uncovered that the ZAR1 resistosome is a calcium-permeable channel, and it is a reference that can been used to elucidate how the G2EPs trigger plant immune responses [47,70]. The acidic residues, Glu and Asp, which were included in the ED motifs of the G2EPs, contribute greatly to the formation of carboxylate for conducting cations across membranes in the ZAR1 resistosome. Meanwhile, acidic residues are also required in order for an E protein to trigger cell death in tobacco and rice. We presume that the G2EPs are formed ion channels in membranes.

Brg11, a TALE-like effector from *Ralstonia solanacearum*, induces truncated, highly active arginine decarboxylase (ADC) mRNAs to elevate the polyamine levels. The increase in the polyamine levels triggers a defense response, shapes the composition from the surrounding microbiome, and gives the pathogen a competitive advantage [71]. The novel virulence mechanism for Brg11 uncovered a ternary microbe–host–microbe interaction model, and it also implies that *E* genes are not the only TALE targets to stimulate the defense response. Niche-enhancing genes are probably another useful resource for plant disease control.

## Figures and Tables

**Figure 1 ijms-23-01524-f001:**
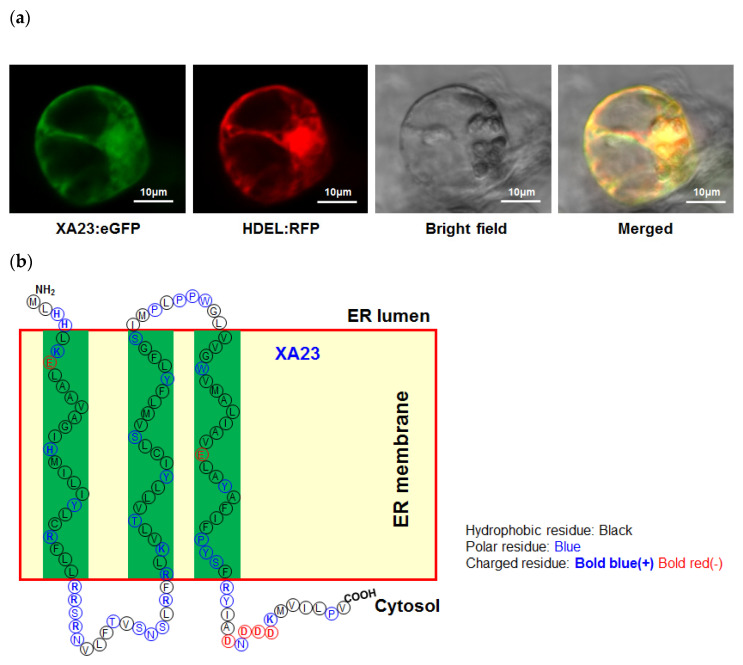
Structural properties for Group 2 E proteins: (**a**) XA23:eGFP and ER marker, HDEL:RFP, colocalize to the endoplasmic reticulum (ER) in rice protoplasts; (**b**) predicted topography of XA23 on the ER membrane; (**c**) structural predictions for Group 2 E proteins. Transmembrane helices predicted by the SOSUI program (https://harrier.nagahama-i-bio.ac.jp/sosui/cgi-bin/msosui.cgi, acceded on 29 November 2021) are underlined. Acidic residues in the predicted ED motif are in bold font. Partially conserved Leu (L) and Ile (I) residues of the hypothetical L(I)-X(4-9)-L(I) motif are in red.

**Table 1 ijms-23-01524-t001:** Basic features of known E proteins from rice and pepper.

	Name	Length ^a^	M.W. ^b^	TM ^c^	HR in *N. benthamiana*	Source and Refernce
Rice	XA7	113	11.8	2	Yes	*Oryza sativa*, DV85 [9,10,11]
	XA10	126	13.9	4	Yes	*Oryza sativa*, Cas 209 [7]
	XA23	113	13.1	3	Yes	*Oryza rufipogon* [11]
	XA27	113	12.1	3	No	*Oryza minuta*, Acc. 101141 [4]
Pepper	BS3	342	37.6	0	Yes	*Capsicum annuum*, ECW-30R [5]
	BS4C-R	164	19.4	4	Yes	*Capsicum pubescen*, PI 235047 [6]

Length **^a^**: number of amino acids; M.W. **^b^**: molecular weight (kilodalton); TM **^c^**: number of predicted transmembrane helices.

## Data Availability

The data presented in this study are available on request from the corresponding author.

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
