# Peer review of "Plant Executor Genes"

_ijms, 2022, doi:10.3390/ijms23031524_

Round 1

Reviewer 1 Report

The review paper titled "Plant Executor Genes" gives a good overview of this fairly new and promising type of plant resistance genes. They recognise the action of TAL -effectors by having an effector-binding element (EBE) in their promotors. Their subsequent expression leads to a hypersensitive response and prevents pathogen growth and disease development.

I found the review very interesting and nicely describes these genes that have been found so far in rice, peppers and recently in barley.

The review is well organised and brings background in the introductory chapter, followed by a chapter on known interactions between the executor genes and TAL -effectors. The next two chapters describe known executor genes from rice and pepper. They also describe likely control strategies for plant diseases that could be used based on what is known about TAL -effectors and their susceptibility genes targets and, of course, executor genes. They wrap up the paper with the discussion.

In my opinion, the article provides a good overview to the interested readers. I have only found one similar review published in 2015 (https://www.frontiersin.org/articles/10.3389/fpls.2015.00641/full). This review additionally summarises the work done in the last six years. Therefore, I recommend the acceptance of the paper.

The only criticism I have is in Figure 1: it looks like box ( c) might be missing in my version of the paper. Perhaps the protein sequences that follow the legend of Figure 1 are part of this c) panel? Although it should show structural predictions, these are just sequences.

Reviewer 2 Report

This manuscript requires very extensive editing of English language. It suffers of endless grammar, sentence structure, and punctuation errors, which make this manuscript hard to read and understand. In particular, I would suggest Authors to avoid complex sentences whenever it is possible and seek a help of a native English language speaker.

Minor comments: sources of information provided in Figures and tables should be  referenced; Figure 1 caption should be placed below the figure (it is in the middle now).

My recommendation is to reject this manuscript and encourage re-submission after extensive language editing

Round 2

Reviewer 2 Report

In the v. 2, the quality of English language has been improved to the level that it does not hinder understanding of the scientific content of this manuscript any more. However, a large number of small errors still remains. As example, I provide a list of errors for the first part of this manuscript, which I noticed at the 1st reading, in a separate file. Since spelling and grammar checking is not a reviewer's responsibility, I did not do this work for the entire manuscript. At this this stage, an English language editing service can be very helpful.

My recommendation is to accept this manuscript for publishing after a moderate English language editing.
